# Eligibility for hepatitis B antiviral therapy among adults in the general population in Zambia

Michael J. Vinikoor[1,2,3]*, Edford Sinkala[2,4], Annie Kanunga[2], Mutinta Muchimba[2], Arianna Zanolini[5], Michael Saag[1], Jake Pry[3,6], Bright Nsokolo[2], Tina Chisenga[7], Paul Kelly[2,8]

**1** Division of Infectious Diseases, University of Alabama at Birmingham, Birmingham, Alabama, United States of America, **2** Tropical Gastroenterology and Nutrition Group, University of Zambia School of Medicine, Lusaka, Zambia, **3** Centre for Infectious Disease Research in Zambia, Lusaka, Zambia, **4** Department of Medicine, University Teaching Hospital, Lusaka, Zambia, **5** Department for International Development, Dar Es Salaam, Tanzania, **6** University of California at Davis, Davis, California, United States of America, **7** Zambian Ministry of Health, Lusaka, Zambia, **8** Barts and the London School of Medicine and Dentistry, Queen Mary University of London, London, United Kingdom

\* mjv3@uab.edu

**Data Availability Statement:** All relevant data are uploaded to figshare (https://doi.org/10.6084/m9.figshare.10184192.v1).

## Abstract

### Introduction

We evaluated antiviral therapy (AVT) eligibility in a population-based sample of adults with chronic hepatitis B virus (HBV) infection in Zambia.

### Materials and methods

Using a household survey, adults (18+ years) were tested for hepatitis B surface antigen (HBsAg). Sociodemographic correlates of HBsAg-positivity were identified with multivariable regression. HBsAg-positive individuals were referred to a central hospital for physical examination, elastography, and phlebotomy for HBV DNA, hepatitis B e antigen, serum transaminases, platelet count, and HIV-1/2 antibody. We determined the proportion of HBV monoinfected adults eligible for antiviral therapy (AVT) based on European Association for the Study of the Liver (EASL) 2017 guidelines. We also evaluated the performance of two alternative criteria developed for use in sub-Saharan Africa, the World Health Organization (WHO) and Treat-B guidelines.

### Results

Across 12 urban and 4 rural communities, 4,961 adults (62.9% female) were tested and 182 (3.7%) were HBsAg-positive, 80% of whom attended hospital follow-up. HBsAg-positivity was higher among men (adjusted odds ratio [AOR], 1.37; 95% confidence interval [CI], 0.99–1.87) and with decreasing income (AOR, 0.89 per household asset; 95% CI, 0.81–0.98). Trends toward higher HBsAg-positivity were also seen at ages 30–39 years (AOR, 2.11; 95% CI, 0.96–4.63) and among pregnant women (AOR, 1.74; 95% CI, 0.93–3.25). Among HBV monoinfected individuals (i.e., HIV-negative) evaluated for AVT, median age

**Funding:** MJV received research support from Gilead Sciences (IN-US-174-3939; https://www.gilead.com) and the Fogarty International Center at US National Institutes of Health (K01TW009998; https://www.fic.nih.gov). The funders had no role in the study design, data collection and analysis, decision to publish, or preparation of the manuscript.

**Competing interests:** MJV received research support from Gilead Sciences (IN-US-174-3939) and the Fogarty International Center at US National Institutes of Health (K01TW009998). The funders had no role in the study design, data collection and analysis, decision to publish, or preparation of the manuscript. Funding from Gilead Sciences does not alter our adherence to PLOS ONE policies on sharing data and materials.

was 31 years, 24.6% were HBeAg-positive, and 27.9% had HBV DNA >2,000 IU/ml. AVT-eligibility was 17.0% by EASL, 10.2% by WHO, and 31.1% by Treat-B. Men had increased odds of eligibility. WHO (area under the receiver operating curve [AUROC], 0.68) and Treat-B criteria (AUROC, 0.76) had modest accuracy. Fourteen percent of HBsAg-positive individuals were HIV coinfection, and most coinfected individuals were taking tenofovir-containing antiretroviral therapy (ART).

## Conclusion

Approximately 1 in 6 HBV monoinfected adults in the general population in Zambia may be AVT-eligible. Men should be a major focus of hepatitis B diagnosis and treatment. Further development and evaluation of HBV treatment criteria for resource-limited settings is needed. In settings with overlapping HIV and HBV epidemics, scale-up of ART has contributed towards hepatitis B elimination.

## Introduction

Globally viral hepatitis is a leading cause of mortality, with most of the deaths occurring in low and middle-income countries (LMIC) due to hepatitis B virus (HBV) [1]. Ambitious global targets have been established to eliminate hepatitis, including a 30% reduction in HBV incidence and treatment of 5 million HBV-infected individuals with antiviral therapy (AVT) by 2020 [2]. Along with Asia, Africa has a large burden of HBV, with an estimated 60 million individuals with chronic HBV infection, >95% of whom are undiagnosed [3]. Major efforts are underway in Africa to raise awareness and encourage governments to adopt and implement policies that will contribute to global hepatitis elimination. The World Health Organization (WHO) has developed and published HBV treatment [4], testing [5], and surveillance guidelines for low and middle-income countries (LMIC). The most common AVT for HBV in Africa is tenofovir, an antiretroviral agent used for millions of HIV-positive individuals, which also has high potency against HBV. There is also a rapidly expanding pipeline of new therapeutic agents to achieve HBV functional or virological cure [6].

In Africa, addressing the gap in high-quality data on HBV has been identified as a major priority [2]. This dearth of data is exemplified in a recent systematic review that identified only 1 recent study from Africa on mother-to-child HBV transmission, the main mode of transmission [7, 8]. Data on AVT for HBV in Africa are particularly scarce and needed because differences in HBV genotypes [9] and host genetics may influence when and who to treat with AVT and treatment outcomes. Young men of African descent may have increased risk of HBV-related hepatocellular carcinoma (HCC) [10]; however, the effect of AVT on HCC risk has not been studied in this population and is inferred from data related to prevention of mother to child transmission, most of which was from Asia.

Several important projects in Africa have helped to address gaps in HBV data and validate international recommendations. In the Gambia, PROLIFICA demonstrated the feasibility and effectiveness of community-based screening, linkage to liver evaluation, and provision of AVT [11]. With a convenience sample of men who have sex with men and prisoners in West Africa and a hospital-based cohort in Ethiopia, investigators demonstrated comprehensive evaluation of HBV monoinfection based using the WHO-recommended approach [12, 13]. Zambia [14] and Tanzania [15] integrated hepatitis B surface antigen testing within a large HIV surveillance

initiative. As HBsAg screening becomes more widespread, new questions are emerging around what proportion of newly diagnosed HBV patients will need AVT. In the only large community-based study to assess this, only 4.4% met criteria for AVT [11]. There is also need to develop and evaluate criteria for AVT in Africa. Recently, the WHO criteria failed to detect around half of Ethiopians with HBV monoinfected patients who were eligible by European guidelines. We performed a population-level survey to evaluate AVT eligibility among HBsAg-positive adults in the general population in Zambia and to assess sociodemographic risk factors for HBsAg-positivity. We also compared the performance of several AVT eligibility guidelines.

## Methods

### Survey sampling frame

We performed a population-based survey, adapted from methods used in the 2013–14 Zambia Demographic and Health Survey (ZDHS) [16]. In Lusaka Province, using Ministry of Health records, we enumerated primary health centers, which have unique and well-defined catchment areas and populations. Among 67 facilities (40 rural and 27 urban), 18 (27%) were selected with probability of selection according to catchment population. Four facilities were rural and twelve were urban, two of which were chosen twice. After sensitizing facility and community leadership, we selected one health facility zone per area at random (each facility has 5–10 defined zones with approximately equal population). From a central landmark, we spun a bottle to choose a direction and in a linear approach from that landmark traveling outward, we conducted door-to-door sampling. Every 3rd household encountered, defined as a dwelling where people currently lived, was eligible for inclusion in the survey, until 100 households per zone were included.

At each participating household, all adult residents ($\geq$18 years) were eligible to participate. We excluded children because Zambian data suggested that HBV immunization, introduced in 2006 for infants aged 6, 10, and 14 weeks, has reduced HBsAg-positivity to 1.3% at ages 0–14 and to 0.7% among those for aged 0–5 [14]. HBV immunization is extremely rare among adults. If no adult was home, 2 additional attempts were made to contact inhabitants, after which another house was added in accordance with the systematic sampling scheme. Written informed consent was obtained before any study-related procedures. This study received ethical approval from the Biomedical Research Ethics Committee at University of Zambia (Lusaka, Zambia) and the Institutional Review Board at University of Alabama at Birmingham (Birmingham, USA), as well authorization from the National Health Research Authority.

### Study measures

All participants had a finger prick point-of-care (POC) hepatitis B surface antigen test (Determine HBsAg, Alere) and completed a brief sociodemographic questionnaire. The questionnaire included a 10-item household possession index that we used to assess income. HBV infection was assumed to be chronic based on a single positive HBsAg result. HIV testing was available upon request but was not a mandatory part of the community survey as there was concern that it would diminish interest in participation. HBsAg (and HIV antibody) results were provided immediately to participants and HBsAg-positives were referred to University Teaching Hospital (UTH) in central Lusaka. To augment the sample with evaluation for AVT, we also included data from 18 HBV monoinfected adults who were diagnosed during the recent ZamPHIA survey in Lusaka. ZamPHIA had similar sampling and recruitment methods. At UTH, additional consent was obtained. HIV-1/2 rapid testing was performed. Among HIV-positives, we documented self-reported current use of antiretroviral therapy (ART). A

clinician performed a physical examination and we collected data on comorbid conditions including diabetes, cancer, and hazardous alcohol use, based on the Alcohol Use Disorders Identification Test-Consumption (AUDIT-C) [17]. We measured ALT, AST, platelets, HBeAg (ETI-EBK Plus, Diasorin, Italy), and HBV DNA, quantified with an in-house assay [18]. HBV genotyping was not performed. AST-to-platelet ratio index (APRI) was calculated [19]. Transient elastography (Fibroscan 402, Echosens, France) was used to measure liver stiffness, a non-invasive test for fibrosis/cirrhosis. In HIV-HBV coinfected individuals, we also measured HIV RNA using an automated platform (Roche COBAS Ampliprep/Taqman) and defined HIV RNA suppression as <1,000 copies/ml. Bus fare was reimbursed for HBsAg-positives who attended the hospital evaluation. After receiving results, patients were referred to the appropriate clinics at either UTH or closer to the participant's home.

## Statistical analysis

We described the proportion of eligible households that participated and the number of individuals tested for HBV. Using bivariable and multivariable logistic regression, we assessed the demographic correlates of HBsAg-positivity including age (categorized as 18–19, 20–29, 30–39, 40–49, 50–59, and 60+ years), sex, pregnancy (pregnant versus not pregnant), income (per asset), marital status (ever versus never married), and location (urban versus rural). Confounders included in the final model were identified *a priori*. Interaction between age and sex, and sex and income were considered for inclusion using a cutoff value of 0.2 in bivarable analysis. We defined linkage to care as the proportion of HBsAg-positives diagnosed in the community survey who attended the clinical evaluation at UTH divided by the total diagnosed in the community. We described the proportion of those assessed at UTH with cirrhosis, based on either ascites on physical examination, liver stiffness measurement (based on transient elastography) ≥9.5 kPa [20], or AST-to-platelet ratio (APRI) >2.0 [19]. ALT elevation was defined as >19 U/L in women and >30 in men.

In each HBV monoinfected individual we applied the EASL 2017 guidelines [21], which we considered the reference standard in this evaluation, and two other HBV guidelines that were specifically developed for LMIC settings: the WHO 2015 guidelines [4], which are largely based on non-African data, and TREAT-B [22], a scoring system developed and validated with African data. EASL 2017 criteria were as follows: cirrhosis based on LSM >9.5 kPa, significant fibrosis based on LSM ≥7.9 kPa and HBV DNA ≥2,000 IU/ml, ALT >80 U/L and HBV DNA >20,000 IU/ml, Metavir ≥A2 on liver biopsy and HBV DNA > 2,000 IU/ml, HBeAg-positivity and age ≥30 years, or family history of HCC or cirrhosis.[21] WHO 2015 criteria were one of the following: clinical cirrhosis (defined as signs of decompensated cirrhosis on physical examination), APRI ≥2.0, or the combination of HBV DNA ≥20,000 IU/ml, ALT elevation, and age ≥30 years. [4]. The TREAT-B criterion is based on categorized ALT level (<20, 20–39, 40–79, and 80+ U/L) and HBeAg status and was validated against EASL [22]. The diagnostic performance of WHO and Treat-B in our patients were compared to EASL 2017, including the sensitivity, specificity, and area under the receiver operating curve (AUROC). All statistical analyses were performed using Stata (Statacorp, USA).

## Results

During June 2017-November 2018, we performed the HBV survey within 16 health facility catchment areas (4 rural and 12 urban) in Lusaka Province. We selected 2,173 households for inclusion in the study, and 1,765 (81.2%) participated. A total of 4,966 adults participated (an average of 2.8 per household; Fig 1) including 1,019 (20.5%) from rural areas. Median age was 32 years (interquartile range, 24–44), 3,118 (62.9%) were women, of whom 217 (7.0%) were

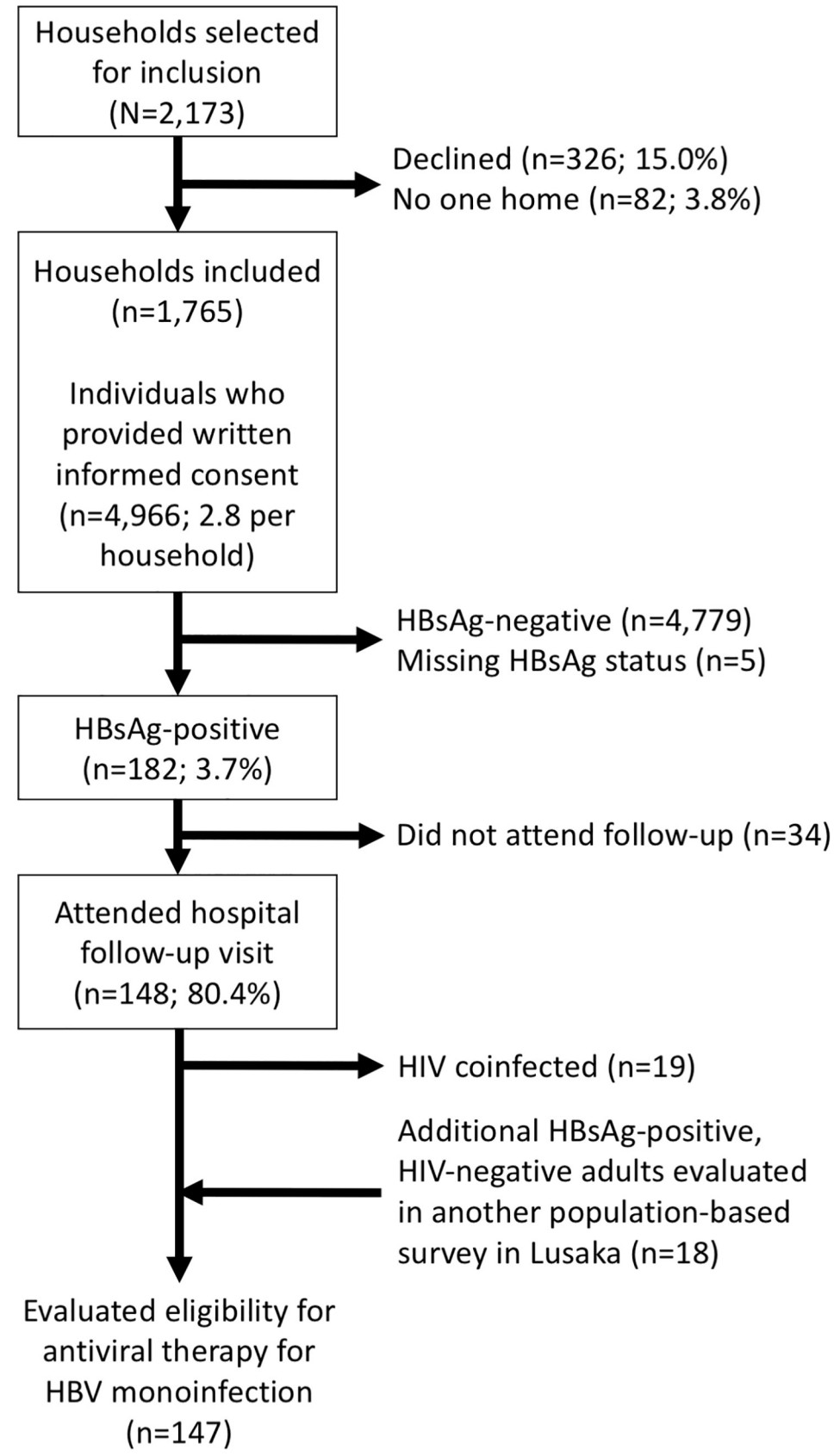

**Fig 1. Recruitment flow diagram.**

pregnant, and 53.4% had at least some secondary education. None of the participants reported prior knowledge of HBV status.

Among the 4,962 with results (4 results were missing), 182 tested HBsAg-positive (prevalence, 3.7%; 95% confidence interval [CI], 3.2–4.2%). No HBsAg-positive individual reported signs or symptoms of acute infection. HBsAg-positivity ranged from 0.3% to 6.1% across the 16 communities and varied somewhat with age category. For example, HBsAg-positivity was 2.6% at ages 18–19 years, 5.4% at 30–39 years, and 0.7% at 60+ years (Table 1). In bivariable analysis, HBsAg-positivity was slightly higher in men compared to women (4.2% versus 3.3%; $P = 0.12$) and among pregnant versus non-pregnant women (5.5% versus 3.0%; $P = 0.04$). Interactions between age and sex and sex and income were not statistically significant. In multivariable analysis, male sex (adjusted odds ratio [AOR], 1.37; 95% CI, 0.99–1.89) was associated with increased HBsAg-positivity and increasing income (AOR, 0.89; 95% CI, 0.81–0.98) was associated with reduced HBsAg-positivity. Also, trends towards increased HBsAg-positivity were seen at ages 30–39 (2.11 times the odds of HBsAg-positivity (95% CI, 0.96–4.63) and among pregnant women (AOR, 1.74; 95% CI, 0.93–3.25). Location (urban versus rural) and marital status were not associated with HBsAg-positivity.

HIV coinfection was relatively common, with 26 (14.4%) HBsAg-positive individuals being HIV-positive. Among the HBsAg-positives diagnosed in the community, 148 (80.9%) were linked to further evaluation at the hospital. We did not document the reasons for failure to link to care. Combining these 148 with 18 HBV monoinfected patients who were also referred

**Table 1. Demographic correlates of HBsAg-positivity.**

|  | Unadjusted OR (95% CI) | Adjusted OR (95% CI) |
|---|---|---|
| Age, in years |  |  |
| 18–19 | Reference | Reference |
| 20–29 | 1.43 (0.73–2.80) | 1.42 (0.70–2.87) |
| 30–39 | 2.16 (1.10–4.25) | 2.11 (0.96–4.63) |
| 40–49 | 1.51 (0.73–3.12) | 1.44 (0.61–3.38) |
| 50–59 | 1.03 (0.44–2.48) | 1.06 (0.41–2.78) |
| ≥60 | 0.26 (0.07–0.95) | 0.26 (0.06–1.05) |
| Sex |  |  |
| Female | Reference | Reference |
| Male | 1.26 (0.94–1.71) | 1.37 (0.99–1.89) |
| Currently pregnant |  |  |
| No | Reference | Reference |
| Yes | 1.62 (0.89–2.96) | 1.74 (0.93–3.25) |
| Marital status |  |  |
| Never married | Reference | Reference |
| Ever married | 1.09 (0.78–1.53) | 0.99 (0.63–1.53) |
| Household income, per 1 item* | 0.90 (0.82–0.98) | 0.89 (0.81–0.98) |
| Location of current residence |  |  |
| Rural | Reference | Reference |
| Urban | 0.82 (0.58–1.17) | 0.74 (0.52–1.06) |

*Based on a 10-item household possession index;

Abbreviations: OR, odds ratio; CI, confidence interval

**Table 2. Characteristics of Zambian adults with HBV monoinfection, by sex.**

| | Overall (N = 147) | Men (n = 71) | Women (n = 76) | P[c] |
|---|---|---|---|---|
| Median age (IQR) | 31 (25–38) | 30 (25–39) | 32 (26–38) | 0.68 |
| Pregnant | 8 (5.4%) | — | 8 (10.5%) | — |
| Median body mass index (IQR) | 22.2 (19.7–25.9) | 21.2 (19.4–23.2) | 24.3 (20.5–27.7) | <0.01 |
| Unhealthy alcohol use[a] | 48 (32.8%) | 32 (45.7%) | 16 (21.0%) | <0.01 |
| Hepatitis B e antigen positive | 34 (24.6%) | 21 (31.3%) | 13 (18.3%) | 0.08 |
| Median ALT, U/L | 25 (17–37) | 32 (23–43) | 20 (14–28) | <0.01 |
| Median AST, U/L | 35 (26–51) | 44 (33–65) | 28 (23–38) | <0.01 |
| ALT elevation[b] | 76 (52.8%) | 38 (54.3%) | 38 (51.4%) | 0.72 |
| Median platelet count, x $10^9$/L | 207 (163–274) | 186 (145–239) | 230 (182–284) | 0.02 |
| APRI score >2.0 | 4 (3.0) | 4 (6.2) | 0 | 0.07 |
| Median liver stiffness, kPa | 6.1 (4.9–8.1) | 6.0 (4.6–6.9) | 6.3 (5.4–8.3) | 0.24 |
| Liver stiffness, n (%) | | | | 0.12 |
| <7.9 kPa | 35 (71.4) | 18 (62.1) | 17 (85.0) | |
| 7.9–9.5 kPa | 9 (18.4) | 8 (27.6) | 1 (5.0) | |
| ≥9.5 kPa | 5 (10.2) | 3 (10.3) | 2 (10.0) | |
| Median HBV DNA $\log_{10}$ IU/ml | 3.1 (2.2–4.9) | 3.4 (2.4–5.8) | 2.9 (2.0–3.8) | 0.07 |
| HBV DNA level, IU/ml | | | | 0.01 |
| <2,000 | 78 (55.7%) | 32 (48.4%) | 46 (62.2%) | |
| 2,000–20,000 | 23 (16.4%) | 8 (12.1%) | 15 (20.3%) | |
| >20,000 | 39 (27.9%) | 26 (39.4%) | 13 (17.6%) | |
| AVT eligible by EASL 2017 | 25 (17.0%) | 16 (22.5%) | 9 (11.8%) | 0.08 |
| AVT eligible by WHO 2015 | 15 (10.2%) | 13 (18.3%) | 2 (2.6%) | <0.01 |
| AVT eligible by TREAT-B score | 42 (31.1%) | 29 (43.9%) | 13 (18.8%) | <0.01 |

[a]Alcohol use was assessed by the Alcohol Use Disorders Identification Test-Consumption (AUDIT-C) and AUDIT-C score of 3+ for women and 4+ for men was considered unhealthy.

[b]ALT elevation was defined as 30+ U/L for men and 20+ U/L for women;

[c]Statistical comparison between men and women;

Abbreviations: ALT, alanine aminotransferase; HBV, hepatitis B virus; APRI, AST-to-platelet ratio index; AVT, antiviral therapy; WHO, World Health Organization guidelines; EASL, European Association for the Study of the Liver guidelines

to our study site from the ZamPHIA survey, we evaluated a total of 166 HBsAg-positive adults at the hospital.

Among HBV monoinfected adults (n = 147; 19 others were HBV-HIV coinfected), 71 (48.2%) were men, median BMI was 22.2 (interquartile range, 19.7–25.9), and 48 (32.9%) were current hazardous drinkers (Table 2). Elevated ALT was noted in 75 (52.1%) at WHO thresholds, 32 (22.2%) at >40 U/L, and 9 (6.2%) at >80 U/L. On physical examination, no patient had signs (i.e., ascites) of decompensated cirrhosis; however, 8 (5.4%) had cirrhosis by TE (n = 5) or APRI (n = 3). HBeAg-positivity was seen in 34 (24.6%) overall and appeared to be more common in men versus women (31.3% versus 18.3%; P = 0.08). Median HBV DNA was 1,392 IU/ml (IQR, 159–75,416), and 62 (44.3%) had HBV VL >2,000 IU/ml. The percentage with ALT elevation was similar between those who did and did not report hazardous alcohol use (56.2 versus 49.5%; P = 0.44).

The proportion eligible for AVT was 17.0% by EASL 2017 criteria, 10.2% by WHO, and 31.1% by Treat-B (Table 2). Men were more likely than women to need therapy across all three criteria (P<0.01), particularly under WHO criteria (18.3% versus 2.6%; P<0.01). The specific criteria met by HBV monoinfected individuals are displayed in Table 3. Among HBV

**Table 3. Eligibility for antiviral therapy for hepatitis B, by international treatment guidelines.**

| EASL 2017 | | WHO 2015 | | Treat-B | |
|---|---|---|---|---|---|
| **Criteria** | **n** | **Criteria** | **n** | **Score** | **n** |
| Cirrhosis and detectable HBV DNA | 8 | Clinical cirrhosis | 0 | 2 | 29 |
| Significant fibrosis and HBV DNA ≥2,000 IU/ml | 5 | APRI >2.0 | 4 | 3 | 10 |
| ALT ≥80 U/L and HBV DNA ≥20,000 IU/ml | 0 | HBV DNA ≥20,000 IU/ml and ALT elevation and age ≥30 years | 11 | 4 | 6 |
| METAVIR ≥2 and HBV DNA ≥2,000 | 0 | | | | |
| HBeAg-positive and age ≥30 years | 12 | | | | |
| Family history of HCC or cirrhosis | 0 | | | | |
| Subtotal | 25 | Subtotal | 15 | Subtotal | 45 |

Cirrhosis was defined as liver stiffness >9.5 kPa; Significant fibrosis was defined as liver stiffness >7.9 kPa; 'Clinical cirrhosis' was defined as signs of decompensated cirrhosis on physical examination; ALT elevation was defined as 20+ U/L for women and 30+ for men;

Abbreviations: EASL, European Association for the Study of the Liver; WHO, World Health Organization; HBV, hepatitis B virus; ALT, alanine aminotransferase, HBeAg, hepatitis B e antigen, HCC, hepatocellular carcinoma; APRI, AST-to-platelet ratio index

monoinfected individuals, 8 pregnant women were assessed. Of these, none had evidence of cirrhosis, 1 had ALT elevation, 2 met EASL criteria for AVT for their own infection, and none of the remaining 6 would have met criteria for tenofovir therapy for prevention of mother-to-child transmission (i.e., HBV DNA <200,000 IU/ml and HBeAg-negative). Considering EASL 2017 as the standard, the accuracy of WHO was poor (AUROC of 0.68; 95% CI, 0.58–0.78). While WHO criteria had relatively high specificity at 95.6%, sensitivity was very low at 40.0%. Treat-B was slightly better than WHO with an AUROC of 0.76 (95% CI, 0.66–0.86; Table 4).

Among the 19 HIV-HBV coinfected patients assessed in the hospital, 10 (55.6%) had ALT elevation by WHO criteria (compared to 52.1% in HBV monoinfection), 4 (23.5%) were HBeAg-positive, and none had evidence of cirrhosis. Eleven reported being on ART and all reported taking tenofovir-containing ART regimens. Time on ART and data related to engagement in care and ART adherence were not available. Among those taking tenofovir-containing ART, 4 (40.0%) had ALT elevation, 8 of 9 had HIV viral suppression (<40 copies/ml), and 6 of 9 had HBV DNA <2,000 IU/ml.

## Discussion

In a general population sample of Zambian adults born prior to availability of HBV immunization, 3.7% were HBsAg-positive and none had prior awareness of their HBV status. Male sex

**Table 4. Performance of WHO 2015 and Treat-B criteria to classify initial eligibility for antiviral therapy for hepatitis B.**

| Criteria | WHO 2015 | Treat-B |
|---|---|---|
| Sensitivity (95% CI) | 40.0 (21.1–61.3) | 73.9 (51.6–89.8) |
| Specificity (95% CI) | 95.9 (90.7–98.7) | 77.7 (68.8–85.0) |
| Positive predictive value (95% CI) | 66.7 (42.8–84.2) | 40.5 (30.8–50.9) |
| Negative predictive value (95% CI) | 88.6 (85.0–91.5) | 93.6 (87.9–96.7) |
| Positive likelihood ratio | 9.76 | 3.31 |
| Negative likelihood ratio | 0.63 | 0.34 |
| AUROC (95% CI) | 0.68 (0.58–0.78) | 0.76 (0.66–0.86) |

European Association for the Study of the Liver 2017 guidelines were used as the reference standard for this analysis;

Abbreviations: WHO, World Health Organization; CI, confidence interval; AUROC, Area under the receiver operating curve

and lower income were associated with HBsAg-positivity, with trends towards increased HBV among those in their 4th decade and among pregnant women. Based on a hospital-based assessment, 17.0% met EASL 2017 criteria for antiviral therapy. Regardless of the AVT criterion used, men appeared to have increased odds of requiring therapy compared to women. These data can guide implementation of hepatitis elimination activities in Africa.

Our main finding that 1 in 6 HBV monoinfected adults met EASL criteria for HBV AVT at the time of diagnosis was substantially higher than the 4.4% reported from a community sample of adults in the landmark Gambian study PROLIFICA [11]. This difference could be in part due to the change in EASL guidelines from 2015 (used by PROLIFICA) to 2017. However, we also noted that our HBV monoinfected sample had higher proportions with ALT >40 U/L and HBV DNA >2,000 IU/ml than the Gambian sample. Differences in the proportion needing therapy may be due to the sample age, as PROLIFICA recruited a slightly older population (median age of 38 years) or due to HBV genotype, since A1 is more common in Zambia [23, 24] than the Gambia where E predominates, and genotype A1 was linked with more severe disease compared to E [11]. Treatment of HBV in those who need it is one of the 2030 global hepatitis targets [25]; therefore, estimates around the proportion who need therapy are critical to inform global strategies to diagnose and link individuals to care and AVT.

Our study also provides evidence on the limitations of the WHO guidelines for HBV treatment. Compared to EASL 2017, WHO had very low sensitivity to identify HBV monoinfected patients who needed AVT. Among the 25 who met EASL criteria, only 10 (40%) met WHO criteria, similar to a recent study in Ethiopia [26]. Unlike the Ethiopian study, where many patients had complications of HBV at baseline, including decompensated or compensated cirrhosis, our sample was population-based and importantly no patient had decompensated cirrhosis, one of the WHO criterion. We also evaluated the recently developed Treat-B criteria, which rely on ALT, which is widely available in Africa, and HBeAg, which is feasible and could become more available through rapid tests. Treat-B, which had good accuracy during initial validation (AUROC of 0.88), had moderate performance in our sample (AUROC of 0.76), similar to a report from Ethiopia [27].

These data also demonstrate the intersection of the HBV and HIV epidemics in Zambia. Although not assessed in our survey, the ZamPHIA study reported 1.31 times increased odds of HBsAg-positivity among HIV-infected adults [14]. HIV infection accelerates the natural history of HBV infection [28], but in our survey HIV-HBV coinfected patients had similar degree of ALT elevation and HBeAg-positivity to those with HBV monoinfection, likely because of prior/current ART use in coinfected patients. In fact, ART scale-up has already begun to address the HBV treatment gap in Zambia. Per ZamPHIA, there are ~540,000 HBsAg-positive Zambians, including 72,000 with HIV coinfection and 468,000 with HBV monoinfection. Assuming 100% eligibility among coinfected and 17% eligibility among monoinfected individuals (based on EASL), we estimated that ~150,000 need HBV therapy. At present ~60% of HIV-positive Zambian adults are on ART [14], and >90% take a tenofovir-based regimen. Therefore, perhaps half of the 72,000 (i.e., 36,000) HIV-HBV patients may already be on appropriate HBV therapy. This suggests that around 25% of the HBV treatment gap in Zambia has already been addressed by the HIV response. Getting the additional ~120,000 on HBV-active therapy is feasible considering that >800,000 Zambians are currently taking ART for HIV infection.

As HBV elimination strategies are implemented in Africa, these data suggest that men should be a priority population. Based on our data and those from the ZamPHIA study, men have 1.3–1.6 times increased odds of HBsAg-positivity as well as 2–5 times increased odds of needing AVT if HBsAg-positive compared to women. Men were also more likely to be HBeAg-positive in our study, which is a marker of infectivity. HBV-related HCC also tends to

disproportionately impact young African men, though it was not assessed in this study [10]. To reach men, alternative approaches to household testing may be needed, since only ~40% of our survey participants were male, similar to other household surveys. When male-oriented health interventions exist in Africa, such as voluntary medical male circumcision [29], integrating HBsAg testing may be an efficient way to identify men and link them to care.

In our analysis, pregnant women also had a trend toward increased HBsAg compared to non-pregnant women, which could be due to immune suppression that occurs in pregnancy. None of the 8 further evaluated met guidelines for antiviral therapy for PMTCT. Further analysis of this is warranted as our sample size of pregnant women (n = 217) was small.

In our survey, we noted that HBsAg-positivity peaked in the 4th decade of life and there was a trend towards increased odds of HBsAg-positivity among 30- to 39-year-olds compared to 18- to 19-year-olds. This could be due to cohort effects (i.e., changes in behavior or in other public health interventions phased in over time); however, several studies in Africa have suggested that sexual transmission in adulthood may contribute new HBV infections. Sexual risk behavior was linked with HBV infection in Uganda [30], and adults taking HBV-active antiretroviral therapy for HIV had reduced HBV incidence [31]. In homosexual Kenyan men, recent HIV acquisition and rape were associated with higher HBcAb incidence while circumcision was associated with decreased risk [32]. Although preventing vertical and early childhood transmission, when the risk for chronic infection is highest, remains the priority, a better understanding of sexual transmission of HBV in Africa is needed.

Data from this survey have broad relevance to global hepatitis policy. First, these data fill a gap in knowledge around what proportion of adults with chronic HBV infection may need AVT. Secondly, we demonstrated that policymakers and programs need to carefully select the treatment eligibility criteria to be used for HBV monoinfected patients as some criteria are more stringent and require HBV virological testing, something that is rarely available and technically challenging, while others may be low cost but result in over-treatment. In its first national hepatitis guidelines, Zambia has adopted WHO recommendations, and where HBV DNA is not available the decision to initiate AVT is based on persistent ALT elevation [33]. Developing low cost, feasible, and accurate criteria on when to treat chronic HBV in Africa, as well as how long to treat and when to stop, are major priorities. Our study also provides a surveillance approach that could be periodically utilized by governments to assess progress toward hepatitis elimination and to evaluate HBV interventions. These data reinforce that chronic HBV infection in the general population in Africa is usually asymptomatic; none of the HBsAg-positive adults assessed had signs of decompensated cirrhosis. It will be critical for health communication around hepatitis testing to emphasize the indolent nature of the disease, as is done for high blood pressure. Finally, these data can be used to advocate for further integration of HBV and HIV activities in African countries in East and Southern Africa where both infections are common. Achieving the HIV epidemic control targets (i.e., UNAIDS 95-95-95) will also contribute toward hepatitis B elimination in countries with intersecting epidemics.

The major strength of this study was its population-based nature and the inclusion of both urban and rural individuals. This is one of the only population-based studies of its kind for HBV in Africa. Eligibility for AVT is based on the natural history of HBV; therefore, the proportion needing therapy in our sample from Lusaka Province is likely to be a valid estimate throughout Zambia and possibly in other African countries with similar age, sex, and HBV genotype distributions. Our ability to link the majority of community-diagnosed individuals to further evaluation for AVT was another strength. Our study also had weaknesses including lower study uptake by men, which could have created a spurious association between sex and eligibility for antivirals if men who were tested at home were sicker than those not at home.

This is unlikely as HBsAg-positive individuals did not report current symptoms of liver disease and similar association between male sex and meeting EASL criteria was reported in another study [11]. Only a limited number of sociodemographic factors were evaluated. Because of cost and logistical issues, we did not perform HBV genotyping. We did not repeat HBsAg status after 6 months among HBsAg-positives and could have slightly over-estimated the proportion in the population with chronic infection. On the contrary, lack of assessment of occult HBV infection [34] could have lowered our estimate of active HBV infection. Our data would have been strengthened had we collected reasons for failure to link to the hospital component of the study among newly diagnosed HBV patients.

In summary, a larger proportion of HBV monoinfected individuals in the general population in Zambia may require AVT than previously thought. Men were significantly more likely to meet criteria for therapy. Due to its intersection with HBV, the strong HIV response in Zambia has already contributed substantially to hepatitis B elimination.

## Author Contributions

**Conceptualization:** Michael J. Vinikoor, Arianna Zanolini, Paul Kelly.

**Data curation:** Michael J. Vinikoor, Mutinta Muchimba, Arianna Zanolini, Jake Pry.

**Formal analysis:** Michael J. Vinikoor, Arianna Zanolini, Jake Pry.

**Funding acquisition:** Michael J. Vinikoor, Paul Kelly.

**Investigation:** Michael J. Vinikoor, Edford Sinkala, Arianna Zanolini, Michael Saag, Jake Pry, Bright Nsokolo, Tina Chisenga, Paul Kelly.

**Methodology:** Michael J. Vinikoor, Mutinta Muchimba, Arianna Zanolini, Paul Kelly.

**Project administration:** Michael J. Vinikoor, Annie Kanunga, Mutinta Muchimba, Paul Kelly.

**Resources:** Michael J. Vinikoor, Edford Sinkala, Tina Chisenga, Paul Kelly.

**Writing – original draft:** Michael J. Vinikoor, Edford Sinkala, Bright Nsokolo, Paul Kelly.

**Writing – review & editing:** Michael J. Vinikoor, Edford Sinkala, Annie Kanunga, Mutinta Muchimba, Arianna Zanolini, Michael Saag, Jake Pry, Bright Nsokolo, Tina Chisenga, Paul Kelly.

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
