## [Decision Letter · Decision Letter 0]

15 Oct 2019

PONE-D-19-24241

Eligibility for hepatitis B antiviral therapy among adults in the general population in Zambia

PLOS ONE

Dear Dr. Vinikoor,

Thank you for submitting your manuscript to PLOS ONE. After careful consideration, we feel that it has merit but does not fully meet PLOS ONE’s publication criteria as it currently stands. Therefore, we invite you to submit a revised version of the manuscript that addresses the points raised during the review process.

We would appreciate receiving your revised manuscript by Nov 29 2019 11:59PM. To enhance the reproducibility of your results, we recommend that if applicable you deposit your laboratory protocols in protocols.io, where a protocol can be assigned its own identifier (DOI) such that it can be cited independently in the future. For instructions see: http://journals.plos.org/plosone/s/submission-guidelines#loc-laboratory-protocols

We look forward to receiving your revised manuscript.

Kind regards,

Hyang Nina Kim, M.D., M.Sc.

Academic Editor

PLOS ONE

Journal Requirements:

2. Please include additional information regarding the survey or questionnaire used in the study and ensure that you have provided sufficient details that others could replicate the analyses. For instance, if you developed a questionnaire as part of this study and it is not under a copyright more restrictive than CC-BY, please include a copy, in both the original language and English, as Supporting Information. Additionally, if any pre-testing of this questionnaire was performed, include details of the pre-testing population and number of individuals involved. Furthermore, please elucidate your sampling methods and choice of sample size - was a power calculation performed prior to recruitment? Also, post-hoc corrections for multiple comparisons were not reported, please give details or explain why this was not performed.

MJV received research support from Gilead Sciences (IN-US-174-3939; https://www.gilead.com) and the Fogarty International Center at US National Institutes of Health (K01TW009998; https://www.fic.nih.gov). The funders had no role in the study design, data collection and analysis, decision to publish, or preparation of the manuscript.

We note that you received funding from a commercial source: Gilead Sciences

Reviewers' comments:

Reviewer's Responses to Questions

**Comments to the Author**

1. Is the manuscript technically sound, and do the data support the conclusions?

Reviewer #1: Yes

Reviewer #2: Yes

Reviewer #3: Partly

2. Has the statistical analysis been performed appropriately and rigorously? 

Reviewer #1: Yes

Reviewer #2: Yes

Reviewer #3: Yes

3. Have the authors made all data underlying the findings in their manuscript fully available?

Reviewer #1: No

Reviewer #2: Yes

Reviewer #3: No

4. Is the manuscript presented in an intelligible fashion and written in standard English?

Reviewer #1: Yes

Reviewer #2: Yes

Reviewer #3: Yes

5. Review Comments to the Author

Reviewer #1: 1) It is claimed “Men also appeared slightly more likely than women to be HBsAg-positive (4.2% versus 3.3%; adjusted odds ratio, 1.31; 95% CI, 0.96-1.81)”, which is not scientifically / statistically sound since the confidence interval covers the unity.

Reviewer #2: Thank you for the opportunity to review “Eligibility for hepatitis B antiviral therapy among adults in the general population in Zambia.” This manuscript addresses in interesting question about selection of individuals with chronic hepatitis B who are expected to most benefit from HBV treatment. This study provides epidemiology from a well characterized population with chronic hepatitis b (CHB) in Zambia. While the study is similar to other epidemiological studies of CHB, it does highlight regional differences in assessing treatment eligibility and points to the need for further context specific studies to understand regional differences and develop context appropriate treatment criteria. Overall it is well written and uses appropriate statistical methods.

Major comments

1. Use of EASL guidelines is reasonable and comparing a variety of guidelines is helpful. However, this highlights the importance of prospective cohort studies to understand treatment and CHB morbidity risks. While EASL guidelines are reasonable they aren’t validated and involve considerable extrapolation in references to Zambia.

2. Methods: I recommend stating what the African validation was for TREAT-B. That is that is was development of a scoring system to approximate EASL criteria. It was not validated to identify individuals at higher risk for CHB complications.

3. An important finding of this study is current limitations on CHB research; ie the impact on location and genotype context on scores calculated from biomarkers, age, and fibrosis scores. Whether EASL, WHO, or TREAT-B are the most appropriate for this population is actually unknown. In addition, in all of these settings appropriate treatment duration and criteria for treatment discontinuation remain unclear.

Minor comments

1. It is unclear which of the authors is affiliated with 8

2. Incomplete references (eg Shimakawa et al)

Reviewer #3: Comments: Manuscript Number PONE-D-19-24241

The authors have reported on the eligibility for hepatitis B antiviral therapy among adults in the general population in Zambia. The authors may wish to consider the following comments:

1. The authors report, “No HBsAg-positive individual reported signs or symptoms of acute infection”. Well, many newly HBV infected people have no symptoms at all. So, this is not a very rigorous criterion to exclude acute infections. Why was the HBc antibody (IgM anti-HBc) measured in N = 580? Could you please provide some details in your flow chart?

2. “HIV coinfection was common, with 26 (14.4%) HBsAg.” I would suggest that the authors tone this down a bit. The description “relatively high” might more fitting.

3. “Pregnant women were more likely than non-pregnant women to be HBsAg-positivity 5.5% versus 3.0%), but the difference was not statistically significant after adjustment for age (P=0.06)”. The study might be underpowered to make this assessment, (only N = 8 women were pregnant); the result might have been statistically significant with a higher sample size. Besides, P= 0.06 is pretty close to the significance level. I would address this obvious limitation in the text. Moreover, did the authors consider the following covariates: the number of previous deliveries, and hospital stay, which might confound the relationship?

4. An important limitation of the study regarding correlates of HBsAg is missing confounder bias. This should be adequately addressed in the text.

5. The authors mention the higher odds of being HBsAg positive and higher odds of infectivity and state that “Men should be a major focus of HBV elimination strategies in Africa”, but what are the consequences of this for women and children? I am not sure how they propose to do this given the reported gender-based disparity in access to healthcare that already exists in developing countries. The authors address correlates of HBV infection but fail to take marital status into account.

6. The authors state that “The odds of HBsAg-positivity among 30- to 39-year-olds was significantly higher than 18- to 19-year-olds” and give some explanations in this regard including public health interventions etc. But, the birth cohort effect has not been sufficiently addressed here. Sexual transmission and infections including reinfections in adulthood also contribute to this and should be mentioned.

7. Chronic HBV infections are usually asymptomatic so I am not quite sure if I understand what the authors are trying to say in the following sentence: “HBV infection in the general population in Africa is usually asymptomatic as none of the HBsAg positive adults assessed had signs of decompensated cirrhosis”. It is because chronic HBV infection is symptomless, that it is also referred to as a silent epidemic.

8. The authors state, “Income and location (urban versus rural) were not associated with HBsAg”. The authors have not addressed (internal and external) migration patterns in their analysis, specifically in their interpretation. Could this have influenced the recorded prevalence?

9. The authors report “Sociodemographic correlates of HBsAg-positivity were identified”. However, the risk factors that have been considered in the analysis are not comprehensive. This should be addressed in the limitations of the study.

10. I am missing a detailed description of the statistical analysis, including model fit statistics in the methods section. Were interactions looked for?

11. Was multiple-comparison testing done in the adjusted regression models shown in Table 1?

12. What was the R2 of the fully adjusted logistic regression model? How much variation was explained by the fully adjusted logistic regression model?

13. The authors report that “HIV coinfection was noted in 14% of HBsAg-positive individuals and the majority were taking tenofovir-containing antiretroviral therapy (ART).” If this is indeed the case, then what are the ramifications for HBV treatment programs in Africa? In resource-poor settings with overlapping HIV and HBV epidemics one could embed HBV elimination programs in running HIV programs. This should be addressed in brief in the discussion.

14. What are the authors’ recommendations in terms of the available guidelines that should be use in Africa based on their analysis/findings? Some comment in this regard would be helpful.

Minor comments:

Figure 1: In the flowchart the abbreviation “DBS” has been used without the full phrase being used anywhere in the text.

6. PLOS authors have the option to publish the peer review history of their article (what does this mean?). If published, this will include your full peer review and any attached files.

Reviewer #1: No

Reviewer #2: No

Reviewer #3: No

---

## [Decision Letter · Decision Letter 1]

12 Dec 2019

Eligibility for hepatitis B antiviral therapy among adults in the general population in Zambia

PONE-D-19-24241R1

Dear Dr. Vinikoor,

We are pleased to inform you that your manuscript has been judged scientifically suitable for publication and will be formally accepted for publication once it complies with all outstanding technical requirements.

With kind regards,

Hyang Nina Kim, M.D., M.Sc.

Academic Editor

PLOS ONE

Reviewers' comments:

Reviewer's Responses to Questions

**Comments to the Author**

1. If the authors have adequately addressed your comments raised in a previous round of review and you feel that this manuscript is now acceptable for publication, you may indicate that here to bypass the “Comments to the Author” section, enter your conflict of interest statement in the “Confidential to Editor” section, and submit your "Accept" recommendation.

Reviewer #1: All comments have been addressed

Reviewer #3: All comments have been addressed

2. Is the manuscript technically sound, and do the data support the conclusions?

Reviewer #1: Yes

Reviewer #3: Yes

3. Has the statistical analysis been performed appropriately and rigorously? 

Reviewer #1: Yes

Reviewer #3: I Don't Know

4. Have the authors made all data underlying the findings in their manuscript fully available?

Reviewer #1: No

Reviewer #3: Yes

5. Is the manuscript presented in an intelligible fashion and written in standard English?

Reviewer #1: Yes

Reviewer #3: Yes

6. Review Comments to the Author

Reviewer #1: The authors have satisfactorily address all questions; there is no more critique.

The authors have satisfactorily address all questions; there is no more critique.

Reviewer #3: (No Response)

7. PLOS authors have the option to publish the peer review history of their article (what does this mean?). If published, this will include your full peer review and any attached files.

Reviewer #1: No

Reviewer #3: No

---

## [Editor Report · Acceptance letter]

17 Dec 2019

PONE-D-19-24241R1 

Eligibility for hepatitis B antiviral therapy among adults in the general population in Zambia 

Dear Dr. Vinikoor:

I am pleased to inform you that your manuscript has been deemed suitable for publication in PLOS ONE. Congratulations! Your manuscript is now with our production department. 

With kind regards,

on behalf of

Dr. Hyang Nina Kim 

Academic Editor

PLOS ONE